# Tuning of Ag Nanoparticle Properties in Cellulose Nanocrystals/Ag Nanoparticle Hybrid Suspensions by H_2_O_2_ Redox Post-Treatment: The Role of the H_2_O_2_/AgNP Ratio

**DOI:** 10.3390/nano10081559

**Published:** 2020-08-08

**Authors:** Dafne Musino, Camille Rivard, Bruno Novales, Gautier Landrot, Isabelle Capron

**Affiliations:** 1INRAE, BIA, 44316 Nantes, France; dafne.musino@inrae.fr (D.M.); bruno.novales@inrae.fr (B.N.); 2SOLEIL Synchrotron, L’Orme des Merisiers, Gif-sur-Yvette, 91192 Saint-Aubin, France; camille.rivard@synchrotron-soleil.fr (C.R.); gautier.landrot@synchrotron-soleil.fr (G.L.); 3INRAE, TRANSFORM, 44316 Nantes, France

**Keywords:** CNC/AgNP hybrids, H_2_O_2_ redox post-treatment, H_2_O_2_/AgNP mass ratio, oxidation state, XANES-EXAFS

## Abstract

Hybrid nanoparticles involving 10-nm silver nanoparticles (AgNPs) nucleated on unmodified rod-like cellulose nanocrystals (CNCs) were prepared by chemical reduction. H_2_O_2_ used as a post-treatment induced a size-shape transition following a redox mechanism, passing from 10-nm spherical AgNPs to 300-nm triangular or prismatic NPs (AgNPrisms), where CNCs are the only stabilizers for AgNPs and AgNPrisms. We investigated the role of the H_2_O_2_/AgNP mass ratio (α) on AgNPs. At α values above 0.20, the large amount of H_2_O_2_ led to extensive oxidation that produced numerous nucleation points for AgNPrisms on CNCs. On the contrary, for α below 0.20, primary AgNPs are only partially oxidized, releasing a reduced amount of Ag^+^ ions and thus preventing the formation of AgNPrisms and reforming spherical AgNPs. While XRD and EXAFS reveal that the AgNP fcc crystal structure is unaffected by the H_2_O_2_ treatment, the XANES spectra proved that the AgNP–AgNPrism transition is always associated with an increase in the metallic Ag fraction (Ag_0_). In contrast, the formation of new 15-nm spherical AgNPs keeps the initial Ag_0_/Ag^+^ ratio unmodified. For the first time, we introduce a complete guide map for the fully-controlled preparation of aqueous dispersed AgNPs using CNC as a template.

## 1. Introduction

In recent decades, silver nanoparticles (AgNPs) have attracted considerable attention because of their tunable properties, allowing them to be used in various application fields such as sterilization [1,2], catalysis [3,4], electronics [5,6,7], and optics [8,9]. Indeed, AgNP surface plasmon resonance (SPR) properties strongly depend on nanoparticle size and shape (e.g., spheres, rods, cubes, bipyramids, prisms, triangles, and hexagons) [10,11,12], making them suitable for biolabeling [13] or surface-enhanced Raman spectroscopy (SERS) [14,15]. In this context, two-dimensional plate-like nanostructures (also referred to as silver nanoprisms (AgNPrisms) [12]) emerged because of their typical anisotropy (i.e., a lateral dimension larger than the thickness), which allows the tuning of their localized surface plasmon resonance (LSPR) linked to the aspect ratio [12,16,17]. Several approaches have made it possible to synthesize AgNPrisms. Lithographic techniques [17,18] are often used to obtain surface-confined AgNP and AgNPrism arrays with a good size-shape control, but they are not adapted to solution-based applications. Light-mediated methods are based on the use of visible light to orient pre-formed NPs (i.e., photo-induced aggregation [19]). As proposed by Jin et al. [20], monodispersed AgNPrisms with an edge length of 30–120 nm can be obtained via a dual-beam illumination. In addition, photo-induced growth treatment can also be performed on Ag seeds in the presence of Ag^+^ ions that are reduced at the seed surface [21]. Callegari et al. [22] indicate that irradiance conditions can transform AgNPs in suspension into larger NPs with different shapes, demonstrating that the photochemical growth of metallic NPs can be controlled by selecting the light color. Other methods to synthesize AgNPs and AgNPrisms are based on the chemical reduction of silver ions [23,24,25]. Several studies [10,23,25,26,27] propose the use of a reducing agent to produce the Ag seeds (e.g., sodium borohydride, NaBH_4_) and the addition of a capping agent to promote AgNP stabilization (e.g., trisodium citrate (TSC); polyvinylpirrolidine (PVP); and dextran). For example, Sun et al. [28] indicated that Ag triangular nanoplates can be produced from spherical 3.5-nm AgNPs previously synthesized by reducing Ag^+^ ions in the presence of NaBH_4_, PVP, and sodium citrate. The authors underline that PVP provided stable Ag seeds smaller than the critical size necessary for morphological transformation, whereas citrate effectively induced nanoplate formation. However, the extensive use of stabilizers could prevent AgNP and AgNPrism functionalization, preventing them from being used for sensing applications [29]. In addition to capping agents, other authors [10,11,30] propose the introduction of hydrogen peroxide (H_2_O_2_) to promote the formation of AgNPrisms. Indeed, the addition of H_2_O_2_ to the suspension where AgNPs are already synthesized induces the oxidative dissolution of unstable Ag seeds while preserving those with twin defects or stacking defaults [11,30,31]. Moreover, under a neutral environment, H_2_O_2_ is able to reduce Ag^+^ ions into Ag atoms, which then aggregate into seeds necessary for the formation of AgNPrisms. In their pioneering work, Métraux et al. [23] proposed a simple procedure to prepare AgNPrisms in aqueous solution using AgNO_3_/NaBH_4_/PVPV/TSC/H_2_O_2_, controlling the AgNPrisms thickness. In 2011, an approach similar to the one proposed by Métraux et al. was adopted by Zhang et al. [31] to identify the role of each agent involved in AgNPrism formation. They pointed out the critical role of H_2_O_2_ instead of citrate in the well-known chemical reduction route to AgNPrisms. Parnklang et al. [32] developed a novel approach for AgNPrism synthesis with a tunable localized surface plasmon reference, focusing on the chemical shape transformation of AgNPs by the addition of H_2_O_2_ and proving that the H_2_O_2_ injection rate and mixing efficiency are the key parameters to control the LSPR wavelengths.

Finally, biopolymeric templates such as cellulose nanocrystals (CNCs) and nanofibers (CNFs) can also be used as stabilizers in the synthesis of AgNPs and AgNPrisms, allowing the production of new hybrid nanomaterials. These studies [33,34,35,36] used modified CNC and CNF surface chemistry. In this context, 2,2,6,6-tetramethylpyperidine-1-oxyl (TEMPO)-oxidized CNFs appear to be an excellent substrate for the stabilization of AgNPs synthesized by NaBH_4_ reduction since most of the surface primary C6 hydroxyls can be converted to carboxylates by TEMPO oxidation [37]. The introduction of H_2_O_2_ as redox post-treatment allows the conversion from AgNPs to AgNPrisms in this hybrid system as well. To our knowledge, all the post-treatments reported on AgNPs nucleated on CNCs required preliminary surface treatment.

In this study, for the first time, nanocellulose with unmodified surface is used to investigate the impact of a H_2_O_2_ redox post-treatment on the morphology, physicochemical properties, and structural organization of grafted AgNPs. We followed a two-step process: First, AgNP nucleation is initiated by chemical reduction on native CNCs to form a CNC/AgNP hybrid without any additional capping agent and second, a H_2_O_2_ redox post-treatment convert AgNPs into AgNPrisms. We elucidate the role of the initial H_2_O_2_/AgNP mass ratio (α), proposing a detailed characterization of the AgNP-AgNPrism conversion. In particular, we correlate the size-shape variations of AgNPs to their oxidation state evaluated by X-ray absorption near-edge structure (XANES) and to their crystallographic structure using X-ray powder diffraction (XRD) and extended X-ray absorption fine structure (EXAFS). Our approach allows a complete control of the properties of AgNPs in hybrid nanomaterials, opening the way to new application fields.

## 2. Experimental Section

**Chemicals.** Cellulose nanocrystals were purchased from CelluForce (Windsor, Canada, product number 2015-009). They were obtained from bleached Kraft pulp by acid hydrolysis and then neutralized to sodium form and spray-dried (length = 183 ± 88 nm; cross-section = 6 ± 2 nm; aspect ratio = 31) [38]. Silver nitrate (AgNO_3_ ≥ 99%), sodium borohydride (NaBH_4_ ≥ 96%), and hydrogen peroxide (H_2_O_2_) were purchased from Sigma-Aldrich (France) and used without further purification. All of the aqueous suspensions and solutions were prepared using ultra-pure water.

**Synthesis of CNC/AgNP hybrid suspensions and H_2_O_2_ post-reaction.** To produce well-dispersed CNC/AgNP hybrid suspensions, CNC aqueous suspension (2 g/L) was dialyzed against water for 3 days (dialysis bath volume to sample volume = 10:1). Then, 10 mL were mixed at room temperature for 1 min with various amounts of AgNO_3_ aqueous solution (50 mM, from 15 to 700 μL). A quantity of 500 μL of freshly-prepared NaBH_4_ aqueous solution (100 mM) was then added to reduce Ag^+^ ions, obtaining AgNPs. NaBH_4_ aqueous solution was placed in ice to minimize its decomposition. AgNP formation induced a variation of color suspension (i.e., from translucent to yellow). The final suspension was covered with aluminum foil to prevent AgNP oxidation by light, mixed at room temperature for 24 h and then dialyzed against water for 24 h. For the H_2_O_2_ redox post-treatment, various amounts of H_2_O_2_ (0, 40, 80, 120, 160, and 250 μL) were added under stirring to the hybrid suspension immediately after the introduction of NaBH_4_. An exothermic reaction took place, leading to the formation of gas bubbles resulting from the H_2_O_2_ decomposition [10]. Finally, the suspension was dialyzed against water for 24 h to remove unreacted reagents (dialysis bath volume to sample volume = 10:1).

**Characterization.** A Mettler-Toledo UV7 spectrophotometer (Columbus, OH, USA) equipped with a 10-mm quartz cell was used to record the light-visible absorbance of hybrid suspensions in the 300–900 nm range. All the samples were diluted (1:10) and ultra-pure water was used as a blank reference.

The AgNP content in CNC/AgNP hybrid suspensions was determined by digesting 1 mL of sample with 40 mL water/aqua regia mixture (i.e., 30% v aqua regia; HCl/HNO_3_: 3/1) and then analyzing it by atomic absorption spectroscopy (AAS) (ICE 3300 AAS, Thermo Fisher, Waltham, MA, USA). A calibration curve was obtained using a silver standard solution (1000 μg/mL, Chem-Lab NV, Zedelgem, Belgium) at different concentrations, from 0.25 to 10 ppm. Two independent measurements were repeated for each sample. The final AgNP content was expressed in mg of AgNP per g of sample (wt%). 

To obtain scanning transmission electron microscopy (STEM) acquisitions, hybrid suspensions were diluted with water at 0.5 g/L in CNC content. Then, 10 μL were deposited on glow-discharged carbon coated grids (200 meshes, Delta Microscopies, Mauressac, France) for two minutes and the excess was removed using Whatman filter paper. The grids were dried overnight in air and then coated with a 0.5-nm platinum layer by an ion-sputter coater (LEICA AM ACE600, Wetzlar, Germany). Images were recorded with a Quattro S field emission gun scanning microscope (Thermo Fisher Scientific, Waltham, MA, USA) at 10 kV using a STEM detector. The acquired STEM images were analyzed by ImageJ software to determine the mean AgNP Feret diameter (i.e., the largest distance between two tangents to the contour of the measured particle), averaged over the largest possible number of particles (from 20 to 100, depending on the sample). 

XANES and EXAFS measurements were performed to study the AgNP oxidation state and bulk atomic structure (e.g., bond length, interatomic distance), respectively. XANES-EXAFS spectra were simultaneously recorded in transmission mode at the Ag K-edge from 25,250 to 27,750 eV on a SAMBA beamline at the SOLEIL synchrotron (Saint Aubin, France). The Si (220) monochromator was calibrated to 25,515.6 eV at the first inflection point of the Ag foil XANES spectrum. The freeze-dried hybrid samples were pressed to obtain circular pellets with a diameter of 6 mm with a controlled amount of AgNPs to reach an absorption edge jump close to 1. The pellets were placed on a sample rod and immersed in a liquid nitrogen bath before being introduced into the He cryostat (T = 20 K). Silver foil (Agfoil) and AgNO_3_ aqueous solution with 1 wt% glycerol (AgH_2_O) were used as standards. For each sample, one scan was collected in transmission and in continuous scan mode along the 25,250 to 27,750 eV energy range with 5 eV/s monochromator velocity and 0.08 s/point integration time. Scans were normalized and background-subtracted using the Athena software package [39]. XANES data were analyzed by a linear combination fitting (LCF) procedure using the fit range [E_0_ − 20 eV, E_0_ + 50 eV] with E_0_ set to 25,514 eV, and using Agfoil and AgH_2_O standards as components. All component weights were forced to be positive, and the relative proportions of the components were forced to add up to 100%. The EXAFS oscillations were background-subtracted using an autobk algorithm (Rbkb = 1, k-weight = 3) and the Fourier transform of the k^3^-weighted EXAFS spectra was calculated over a k range of 2.5–18 Å^−1^ using a Hanning apodization window (width of the transition region window parameter = 1). k^3^ EXAFS fitting was performed in the 2.35–7.7 Å distance range (R) with the Artemis [39] interface to IFEFFIT library, using least-squares refinements. Paths used for fitting standards and samples were obtained from a metallic silver crystallographic model [40] using the FEFF6 algorithm included in the Artemis interface. Only paths with a rank higher than 7% were considered. E_0_ was fixed to 25520 eV. For sample fitting, the amplitude reduction factor S_0_² was fixed to 0.978 after being determined by fitting the first coordination sphere of the Agfoil spectrum over a range of 2.30–2.83 Å. Degeneracy of the paths, energy shift ΔE_0_, R shift ΔR, and thermal and static disorder σ² were fitted for each of the selected paths for a total of 52 independent points and 19 variables. All R-factors were lower than 0.05.

A Bruker D8 Discover diffractometer was used to record XRD diffractograms. Cu-Kα1 radiation (Cu Kα1,1.5405 Å) was produced in a sealed tube at 40 kV and 40 mA, parallelized using a Gobël mirror parallel optic system and then collimated to produce a 500-mm beam diameter. The data were collected in a 2*θ* angle range from 3° to 70° (10 min of acquisition). The AgNP crystallite size (CS) was determined using Scherrer’s equation [41]:(1)$$\mathrm{CS}=\frac{K\lambda }{\beta cos\theta }$$
where K is the shape factor (0.9), *λ* is the X-ray wavelength (1.54 Å), *β* is the full-width at half-maximum (FWHM), and *θ* is the angle of the diffraction peak of the crystalline phase (Bragg’s angle). The FWHM was determined considering the AgNP characteristic peak at 2*θ* = 38°.

## 3. Results and Discussion

The H_2_O_2_ redox reaction is known to lead to the conversion of spherical AgNPs into AgNPrisms. To shed light on the role of the H_2_O_2_/AgNP mass ratio (α) on the AgNP-AgNPrism transition, we proposed a two-step approach. We first synthesized CNC/AgNP hybrid aqueous suspensions where hydrophilic CNCs are used as substrate to easily disperse and stabilize AgNPs of about 10 nm in water. Indeed, the good CNC dispersion is ensured by the negative surface charges (SO_3_^−^ groups), and the hydroxyl groups on the CNC surface act as nucleation points that allow the growth of well-dispersed AgNPs on the CNC surface [42]. In a second step, the H_2_O_2_ redox post-treatment is performed adding various H_2_O_2_ volumes to the primary hybrid suspension. The following notations will be used from now on: (i)AgNPs: the primary 10-nm spherical NPs nucleated on the CNC surface in initial CNC/AgNP hybrid suspensions;(ii)AgNPs_H_2_O_2_: AgNPs after the addition of H_2_O_2_;(iii)AgNPrism: AgNPs_H_2_O_2_ for which the H_2_O_2_ post-treatment leads to the formation of triangular shaped NPs [12,34].

Such an experimental approach made it possible to investigate the AgNP size-shape transition as a function of α.

Firstly, we focused on a CNC/AgNP hybrid aqueous suspension at 8.7 wt% AgNP treated with different volumes of H_2_O_2_ (0, 40, 80, 120, 160, and 250 µL), thus varying the α parameter from 0 up to 0.42 (Figure 1a). After the addition of H_2_O_2_, the samples were dialyzed and the average amount of AgNP (AgNPs_H_2_O_2_) was estimated at 6.9 ± 1.2 wt% for the six samples, proving the efficiency of the H_2_O_2_ reduction. In Figure 1a, the reference hybrid suspension (i.e., no H_2_O_2_) displays the typical yellow color of aqueous suspensions with well-dispersed and stabilized AgNPs. The increasing addition of H_2_O_2_ enshrines a variation of the color of the suspension from yellow to blue. In the UV-Vis spectra (Figure 1b), the λ_max_ value of the in-plane dipole surface peak was shifted from 400 nm for the reference sample, to 450 nm and 495 nm for samples at α equal to 0.07 and 0.13, respectively. This shift is associated with an increase in the average diameter of AgNPs_H_2_O_2_ from 10 nm for the reference, to 34 nm and 52 nm for α of 0.07 and 0.13, respectively, as measured on STEM images (Figure 1c). Furthermore, the presence of two shoulders around 340 and 380 nm in the UV-Vis spectra reveal the beginning of a morphological modification of the primary AgNPs, even if the AgNPs_H_2_O_2_ still displayed a quite well-defined spherical shape. At α = 0.20, a sharp low intensity peak is present at λ_max_ = 335 nm (red line in Figure 1b), which is generally referred to as the out-of-plane quadrupole resonance peak, representing a good indicator of the AgNP architectural modification related to the aspect ratio [19]. Indeed, the AgNPs_H_2_O_2_ lost their spherical shape, assuming an irregular triangular shape with an increase in their average size (i.e., shift of the in-plane peak to a higher wavelength). The low-intensity peak at 335 nm was also identified in the UV-Vis spectra of hybrids at α equal to 0.27 and 0.42, for which the in-plane dipole plasmon peak was outside of the measurement window. At these high α values, the AgNPs_H_2_O_2_ presented a well-defined triangular shape with an average diameter of up to 324 nm. Such a size-shape transition clearly identified the passage from 10-nm spherical NPs to 300-nm AgNPrisms. We emphasize here that the synthesis of primary AgNPs and the subsequent production of AgNPrisms by H_2_O_2_ redox treatment can be achieved using unmodified native CNCs as stabilizers, without the introduction of additional capping agents, since hydroxyl groups on the CNC surface act as perfect nucleation points [42].

Since silver can exist in various forms (e.g., metallic Ag_0_, ionic Ag^+^, and Ag_2_O oxide), the oxidation state of AgNPs and AgNPs_H_2_O_2_ in hybrid samples were characterized by XANES. An example of fitting of spectra in the XANES region by the LCF procedure was reported in Appendix A. The analysis of the XANES spectra of CNC/AgNP hybrid suspensions at 8.7 wt% AgNP where the α parameter varied from 0 up to 0.42 (Figure 2a) made it possible to reveal the increase in the variation of the Ag_0_ content from 65% to 94% with the increase in the α value. The R-factor and the Chi-square values of the fits of the XANES spectra are reported in Appendix A. 

As for the reference hybrid, the XRD diffractograms of all the CNC/AgNP_H_2_O_2_ (Figure 2b) were characterized by peaks at 38.1°, 44.0°, and 64.2°, corresponding to (111), (200), and (220) planes, respectively. This clearly defined a face-centered cubic (fcc) silver structure, characterized by the isotropic nature of the crystals [34] (JCPDS Card No. 89-3722). In addition to XANES, the EXAFS spectra of the same samples were recorded and analyzed. All Fourier transform spectra of the CNC/AgNP hybrid containing 8.7 wt% AgNP (Appendix A) were fitted with the same crystallographic structure of metallic silver (Ag_0_), with an R-factor systematically lower than 0.016. The overall variation in interatomic distance (R) values obtained from the fits with increasing α from 0 to 0.47 (Appendix A) were systematically negligible as the error bars associated with the R values always partly overlapped with each other. This indicated that the interatomic distances in the hybrids did not significantly change in comparison to the metallic silver distances and that the space group of the AgNPs still corresponded to the fcc silver structure, as suggested by XRD. It therefore follows that the initial and final crystal structural organizations were not affected by the H_2_O_2_ redox post-reaction. These experimental results seemed to suggest that the size-shape transition from AgNPs to AgNPrisms could be achieved only above a critical α value of 0.20, and that the H_2_O_2_ post-reduction did not affect the crystalline structure while partially modifying the oxidation state of the AgNPs_H_2_O_2_. 

To better shed light on the impact of the α parameter on the efficiency of the H_2_O_2_ redox post-treatment and on the morphological and physicochemical properties of AgNPs_H_2_O_2_, hybrid suspensions at a fixed concentration but containing 12.5, 18.6, or 24.7 wt% AgNP were also mixed with various H_2_O_2_ volumes (from 40 to 250 μL). All the α values considered in the study are summarized in Table 1. After introducing H_2_O_2_, the Ag contents were found to be equal to 9.3, 17.1, and 24.1 wt%, respectively, confirming once again that this redox post-treatment occurs at a very high yield, close to 100%. In Figure 3a, we propose STEM images of hybrids at various initial AgNP contents mixed with 160 µL of H_2_O_2_, thus varying α from 0.09 to 0.27. It appeared that the formation of AgNPrisms with a size of 150–300 nm was obtained only at α equal to 0.20 and 0.27. In contrast, AgNPs_H_2_O_2_ produced at α equal to 0.09 and 0.12 maintained a spherical shape with an average diameter of 15–20 nm. AgNP_H_2_O_2_ size distributions are reported in Appendix A, and the NP average diameters are summarized in Appendix A. These last results agreed with those at α equal to 0.12 (18.6 wt% AgNP) and 0.09 (24.7 wt% AgNP), we found the same spherical morphology as that obtained for hybrids at α equal to 0.13 and 0.07 (8.7 wt% and 12.5 wt% AgNP, respectively). On the other hand, AgNPrisms were formed for higher α, as shown at 0.20 and 0.27. Furthermore, it could be observed that smaller satellite AgNPs (between 10 and 35 nm) were formed near AgNPrisms, which were not observed before the H_2_O_2_ reduction step (Figure 1c and Figure 3a). Even if the average diameter of these smaller AgNPs was reported in Appendix A, they were not considered to determine the average size of the AgNPrisms.

The UV-Vis spectra (Figure 3b) showed the shift of the main peak outside of the measurement window and the appearance of the low-intensity peak at λ_max_ = 335 nm only for hybrids at α of 0.20 and 0.27, indicating a size-shape transition of 10-nm spherical AgNPs to 150–350 nm AgNPrisms. For other hybrids, the in-plane peak remained at around λ_max_ = 445 nm, with a wide low-intensity shoulder between 345 nm and 375 nm. In this case, AgNP_H_2_O_2_ maintained a spherical shape grafted on CNC with a slight increase in the average diameter from 10 nm to 15–20 nm.

On the basis of these results, we propose a H_2_O_2_ redox mechanism where H_2_O_2_ first induced the oxidative dissolution of primary AgNPs, generating Ag^+^ ions. After that, two scenarios are occurring; at α equal to or greater than 0.20 (i.e., hybrids at lower initial AgNP contents), the H_2_O_2_ oxidation affected quickly most of the AgNPs, except the contact location where AgNPs are effectively grafted onto the CNC surface. Indeed, H_2_O_2_ etching usually works on the most unstable NPs, while maintaining parts with high stability intact [30,31]. These residual sites on the CNC surface could actually work as nucleation sites for the formation of newly formed AgNPrisms. We assumed that the appearance of the bigger AgNPrisms was promoted by the limited number of Ag residual sites on the CNC surface. On the other hand, for α below 0.20 (i.e., hybrids at higher initial AgNP contents), even the introduction of the largest H_2_O_2_ volume probably induced only a partial etching of most of the primary AgNPs, due to the higher number of NPs with respect to the amount of H_2_O_2_ introduced. Thus, it appeared that the reduction step did not allow the synthesis of AgNPrisms, as in the previous case. The higher number of residual sites available on the CNC surface from the oxidative dissolution could prevent the formation of 300-nm AgNPrisms. A schematic representation of this H_2_O_2_ oxidation-reduction process is proposed in Figure 3c. It could thus be concluded that the mechanism that controlled the size-shape transition of primary AgNPs in a CNC/AgNP hybrid suspension was mainly governed by the H_2_O_2_/AgNP ratio. As for the primary AgNPs, the AgNPrism stabilization was provided by CNCs without any additional stabilizer since, otherwise, AgNP and AgNPrism aggregation and sedimentation occurred. 

As an example, the XANES spectra for hybrids treated with 160 µL of H_2_O_2_ at different initial AgNP contents (α from 0.09 to 0.27) are shown in Figure 4a and the R-factor and Chi-square values of the fits are reported in Appendix A. Extending this analysis, Table 1 reports the evolution of the average diameter of the AgNPs_H_2_O_2_ as a function of their Ag_0_ content for all the studied hybrids (Figure 4b). It is shown that the Ag_0_ content of AgNPs_H_2_O_2_ varied with their morphological modification. Notably, hybrids at 18.6 and 24.7 wt% AgNP_H_2_O_2_ always displayed a α lower than 0.20, maintaining 15-nm AgNPs_H_2_O_2_ with a spherical shape and a rather low Ag_0_ content that remained between 35% and 50%. On the other hand, when the AgNP–AgNPrism transition occurred (i.e., α ≥ 0.20), the Ag_0_ content increased up to 100%. Such a trend confirmed that the size-shape transition from 10-nm AgNPs to 300-nm AgNPrisms was always associated with a modification of the oxidation state. In other words, it could be concluded that it was not possible to obtain AgNPrisms by a H_2_O_2_ redox post-treatment without affecting their oxidation state. Thus, Figure 4b could be considered as a guide map to tune the AgNP morphological characteristics with respect to their oxidation state. Moreover, our experimental results proved that AgNPs and AgNPrisms were undoubtedly composed of different fractions of both Ag_0_ and Ag^+^.

Concerning the structure, the XRD diffractograms showed the persistence of a fcc Ag crystalline structure for all the samples (Appendix A), independently of the α value. Hybrids formulated with the introduction of H_2_O_2_ still showed a well-defined (111) peak, indicating that even bigger AgNPrisms with a triangular shape had a fcc structure and that they were preferentially oriented parallel to the substrate during the acquisition. However, the crystallite size increased from 3.2 nm up to 7 nm with the progressive addition of H_2_O_2_, which confirmed that oxidation/reduction steps occurred and maintained the fcc structure intact, with a slight variation of the CS.

The EXAFS Fourier transform spectra (Appendix A) of the CNC/AgNP hybrid were fitted with the crystallographic structure of metallic silver, with an R-factor systematically lower than 0.048 (Appendix A). The overall variation in interatomic distance values, R, obtained from the fits were systematically negligible as the error bars associated to the R values always partly overlapped with each other. 

This result showed that the interatomic distances in the CNC/AgNP_H_2_O_2_ hybrids do not significantly change in comparison to the metallic silver distances and that the space group of the samples still corresponds to the fcc silver structure, as shown by XRD. As previously shown, the final crystal structural organization was neither affected by the H_2_O_2_ redox post-reaction, nor by the initial Ag content in CNC/AgNP hybrids. 

Finally, we checked the influence of the NaBH_4_/AgNO_3_ initial molar ratio on the H_2_O_2_ redox post-treatment. CNC/AgNP suspensions at the highest silver content (i.e., the lowest NaBH_4_/AgNO_3_ ratio equal to 1.5) were prepared to obtain the NaBH_4_/AgNO_3_ molar ratio of the sample at the lowest AgNP content (i.e., NaBH_4_/AgNO_3_ ratio = 30) and then mixed with 160 μL of H_2_O_2_ (α = 0.09). Even in these conditions, the UV-Vis spectra of such suspensions at an NaBH_4_/AgNO_3_ ratio of 30 overlapped the one of the same sample prepared at a NaBH_4_/AgNO_3_ ratio of 1.5 (Appendix A), thus proving that this parameter did not affect the H_2_O_2_ post-reduction in our experimental conditions. 

## 4. Conclusions

In this study, we investigated the impact of the H_2_O_2_ redox post-treatment on the morphology, physicochemical properties, and structural organization of CNC/AgNP aqueous hybrid suspensions formulated using unmodified CNCs as bio-based support to stabilize AgNPs without the addition of any other capping agent. Hybrids at various AgNP contents (i.e., 8.7, 12.5, 18.6, and 24.7 wt%) were mixed with different H_2_O_2_ volumes (i.e., 0, 40, 80, 120, 160, and 250 µL) to obtain various H_2_O_2_/AgNP mass ratios (α) up to 0.42. 

We demonstrated that a critical α value of 0.20 had to be overcome to achieve a size-shape transition from 10-nm spherical NPs (AgNPs) to 300-nm triangular or prismatic NPs (AgNPrisms). Furthermore, we proposed an H_2_O_2_ redox mechanism that considered the CNCs as stabilizers for AgNP_H_2_O_2_ and AgNPrisms. We speculated that at large amounts of H_2_O_2_, for α values higher than 0.20, the H_2_O_2_ oxidative action concerned most of the NP except its part effectively grafted onto the CNC surface, which acted as a nucleation seed for AgNPrism formation. On the other hand, at α lower than 0.20, primary AgNPs were only partially oxidized and spherical AgNPs-H_2_O_2_ of about 15–20 nm were formed again. 

We proved that the transition from 10-nm spherical AgNPs to 300-nm triangular AgNPrisms (i.e., α ≥ 0.20) was associated with an increase in the Ag_0_ content up to 100%. However, the oxidation state was slightly affected when the H_2_O_2_ post-reaction did not modify the size and shape of AgNPs_H_2_O_2_ (i.e., α < 0.20). Finally, the AgNP_H_2_O_2_ structure was not affected by the H_2_O_2_ redox reaction since a fcc model structure was maintained, regardless of the α value. 

The present results make it possible to create a guide map to fully control AgNP properties in hybrid NPs where CNCs serve as substrate, making these hybrid NPs suitable for new applications fields.

## Figures and Tables

**Figure 1 nanomaterials-10-01559-f001:**
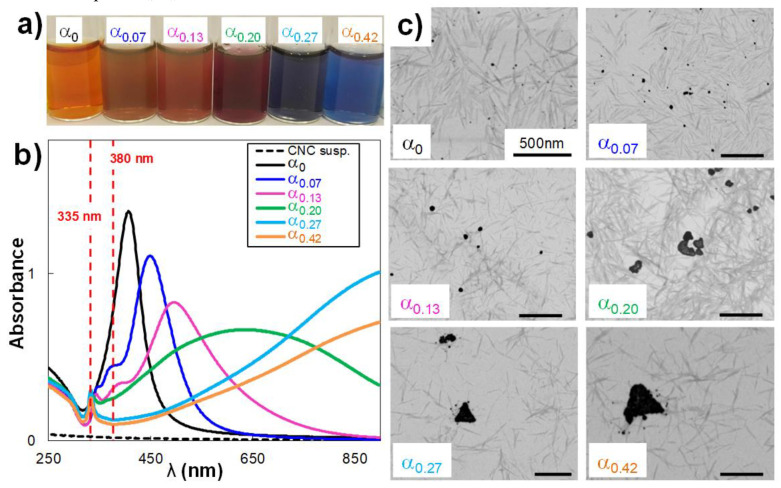
(**a**) color evolution; (**b**) UV-Vis spectra; and (**c**) corresponding scanning transmission electron microscopy (STEM) images of cellulose nanocrystal (CNC)/silver nanoparticle (AgNP) hybrid suspensions at 8.7 wt% AgNP treated with various H_2_O_2_ volumes (i.e., variation of H_2_O_2_/AgNP mass ratio, α). In STEM images, bar scales of 500 nm are indicated.

**Figure 2 nanomaterials-10-01559-f002:**
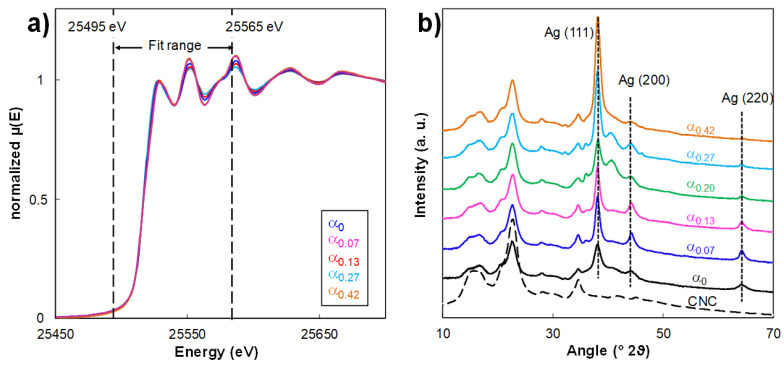
(**a**) X-ray absorption near-edge structure (XANES) spectra and (**b**) XRD diffractograms of CNC/AgNP hybrid suspension at 8.7 wt % AgNP mixed with different amounts of H_2_O_2_ (i.e., various H_2_O_2_/AgNP mass ratios, α).

**Figure 3 nanomaterials-10-01559-f003:**
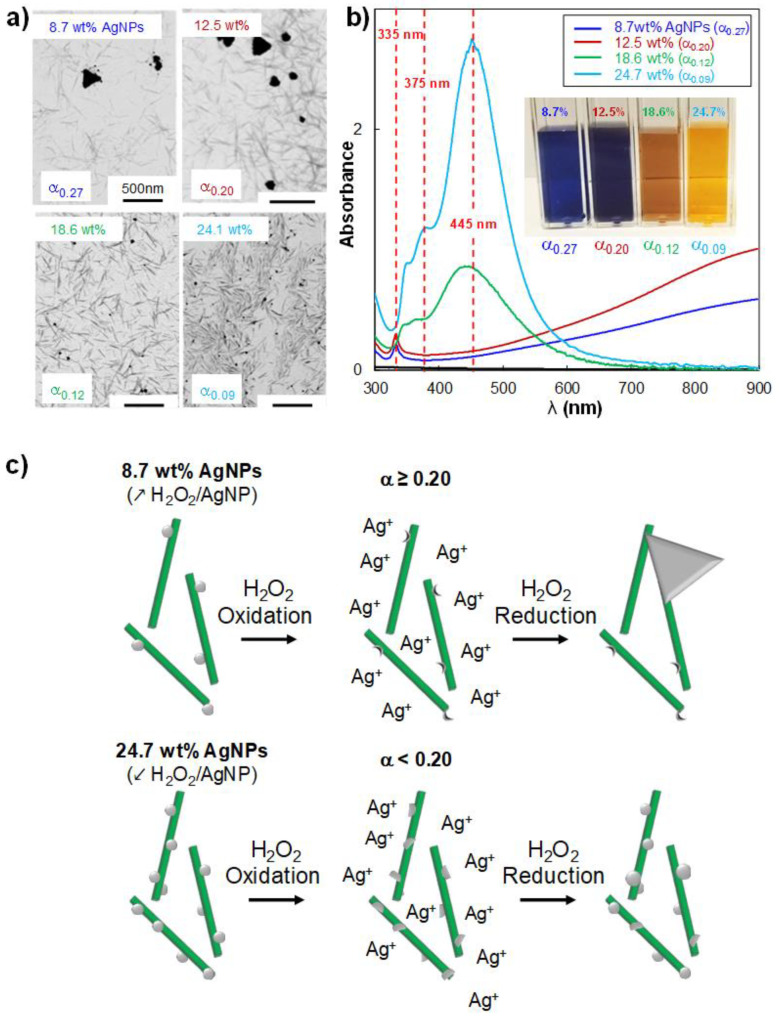
(**a**) STEM images and (**b**) UV-Vis spectra of CNC/AgNP_H_2_O_2_ hybrids at various AgNP_H_2_O_2_ contents where 160 µL of H_2_O_2_ were added. In STEM images, bar scales of 500 nm are indicated. (**c**) schematic representation of the H_2_O_2_ oxidation-reduction mechanism in hybrids at low and high AgNP contents (i.e., α parameter above and below the critical value of 0.20).

**Figure 4 nanomaterials-10-01559-f004:**
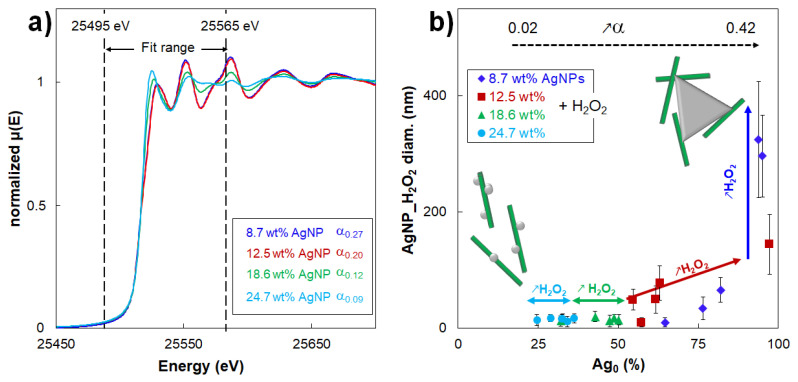
(**a**) XANES spectra of hybrids at different initial AgNP content treated with 160 µL of H_2_O_2_ (i.e., α values above and below critical value of 0.20) (**b**) evolution of the average diameter of AgNPs_H_2_O_2_ in CNC/AgNP_H_2_O_2_ hybrids as a function of their Ag_0_ content estimated from XANES spectra.

**Table 1 nanomaterials-10-01559-t001:** Characteristics of CNC/AgNP hybrids at different AgNP contents with a H_2_O_2_ redox post-treatment (CNC/AgNP_H_2_O_2_).

Ag Content in CNC/AgNP Hybrid (wt%)	H_2_O_2_ Vol. (μL)	Ag Content in CNC/AgNP_H_2_O_2_ Hybrid (wt%)	α (H_2_O_2_/AgNP Mass Ratio)	AgNP_H_2_O_2_ Diameter (nm) ^1^	CS Crystallite Size (nm) ^2^	Ag_0_ Content (%) ^3^
8.7 ± 0.05	0	6.9 ± 1.2	-	10 ± 7	3.1	65 ± 2
	40		0.07	34 ± 20	7.0	77 ± 2
	80		0.13	52 ± 21	7.4	82 ± 2
	120		0.20	121 ± 68	6.5	-
	160		0.27	296 ± 70	6.5	95 ± 3
	250		0.42	325 ± 100	7.0	94 ± 3
12.5 ± 0.08	0	9.3 ± 1.9	-	11 ± 6	3.2	57 ± 2
	40		0.05	49 ± 23	-	62 ± 2
	80		0.09	49 ± 18	-	55 ± 2
	120		0.14	77 ± 29	-	63 ± 2
	160		0.20	145 ± 51	6.7	97 ± 3
	250		0.29	-	-	100 ± 3
18.6 ± 0.1	0	17.1 ± 1.3	-	12 ± 9	-	32 ± 1
	40		0.03	12 ± 10	-	47 ± 1
	80		0.06	17 ± 7	-	49 ± 1
	120		0.09	20 ± 9	-	43 ± 1
	160		0.12	14 ± 9	6.2	50 ± 2
	250		0.18	-	-	41 ± 1
24.7 ± 0.2	0	24.1 ± 1.5	-	11 ± 9	3.0	34 ± 1
	40		0.02	14 ± 10	-	25 ± 1
	80		0.05	16 ± 8	-	33 ± 1
	120		0.07	17 ± 8	-	36 ± 1
	160		0.09	17 ± 6	7.3	29 ± 1
	250		0.15	16 ± 7	-	32 ± 1

**-** Not measured, ^1^ by STEM analysis, ^2^ by XRD, ^3^ by XANES, and the standard error as 3% of the measured value.

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
