# Peer review of "Tuning of Ag Nanoparticle Properties in Cellulose Nanocrystals/Ag Nanoparticle Hybrid Suspensions by H2O2 Redox Post-Treatment: The Role of the H2O2/AgNP Ratio"

_nanomaterials, 2020, doi:10.3390/nano10081559_

Round 1
Reviewer 1 Report
The submitetd manuscript deeply studied the role of H2O2 Redox Post-Treatment on Ag Nanoparticle morphologies when CNC are used as substrate for preparation of AgNPs. In my opinion, the amount of presented results, their nice presentation and the excellent discussion section deserve the publication in its present form. The only suggestion that I have is to move some of the supporting images (most significant) as images in the main text, to better follow the approach and the comments.
Additionally, please revise few lines where the spacing was altered an dthe format of tables (as in the case of Table 1) and modify them.
Author Response
Please, see the attachment.

Reviewer 2 Report
The paper is devoted to a study of cellulose nanocrystals/Ag nanoparticle hybrid suspensions undergoing H2O2 treatment. The topic is interesting and deserves the publication in the Materials journal. However, several points should be clarified before the paper will be suitable for publications.
The authors employ a simple linear combination fit for the analysis of XANES for Ag nanoparticles (Figure S1(b)) using XANES spectra of Ag foil and AgNO3 aqueous solution as components (see page 5 and Figure S1(a)). The validity of such a model has not been proven and should be described in detail. The model ignores the size effect of metallic nanoparticles but, at the same time, introduces a hydrated Ag+ ion environment, which is not present in the Ag-nanoparticles. Why?
The analyses of the Ag K-edge EXAFS spectra should be explained in more detail. The authors used the IFEFFIT approach and performed the fit in a wide k and R-space ranges (FigureS3(b)) based on the fcc Ag crystallographic structure. Since the crystallite size of the nanoparticles is small being below 10 nm (Table 1), it is surprising that no reduction of the interatomic distances (see Montano et al, Phys. Rev. B 30 (1984) 672-677) was observed. Also, the effect of the crystallite size (surface atoms contribution) is not considered in the EXAFS analysis. Why is it being ignored?
The results reported in Table S2 should be reviewed and discussed. Can these figures be trusted? Is there anything useful to be learned from variations in the structural parameters of the model obtained from the fits?
Author Response
Please, see the attachment

Round 2
Reviewer 2 Report
The revision and author's comments are acceptable for the publication of the manuscript.